# Postpartum Follow-Up of Women Who Developed Subclinical Hypothyroidism during Pregnancy

**DOI:** 10.3390/medsci8030029

**Published:** 2020-08-03

**Authors:** Anastasia Linardi, Ioannis Kakoulidis, Ioannis Ilias, Aikaterini Michou, Athina Pappa, Evangelia Venaki, Eftychia Koukkou

**Affiliations:** Department of Endocrinology, Diabetes and Metabolism, Elena Venizelou General and Maternity Hospital, 11521 Athens, Greece; an.linardi@yahoo.gr (A.L.); i_kakoulidis@yahoo.gr (I.K.); katerina.michoy@yahoo.com (A.M.); athpappa@gmail.com (A.P.); evivenaki@gmail.com (E.V.); ekoukkou@gmail.com (E.K.)

**Keywords:** subclinical hypothyroidism, pregnancy, postpartum, lactation, levothyroxine

## Abstract

There is inconsistency in the literature regarding the management of women diagnosed with subclinical hypothyroidism (SCH) during pregnancy in the postpartum period. The purpose of our study was to assess the need for continuation of levothyroxine (LT4) treatment after delivery. We conducted a retrospective cohort study of 114 women with new-onset SCH during pregnancy and at 1-year follow-up postpartum. Criteria for continuation of LT4 after delivery were breastfeeding, thyrotropin (TSH) levels at diagnosis >5 mIU/L, positive antithyroid antibodies and LT4 dose before delivery >50 μg/day. On treatment initiation, mean TSH ± SD was 5.24 ± 2.55 mIU/L. One year after delivery, most patients (86/114) were still on LT4. This was related to TSH levels at the initiation of treatment in gestation (*p* = 0.004) and inversely related to primiparity (*p* = 0.019). In the group of patients who stopped LT4 postpartum, treatment was reinstated in 11 out of 39 (28.2%) due to SCH relapse (mean TSH ± SD = 9.09 ± 5.81 mIU/L). Most women in our study continued treatment after delivery, and a considerable number of women who had discontinued LT4 restarted treatment postpartum. These results stress the need to reassess thyroid function at 6 to 12 months postpartum.

## 1. Introduction

Pregnancy has a profound impact on the thyroid gland and its function, including iodine metabolism [1,2,3,4,5,6,7]. Women with a normal functioning thyroid gland, and adequate iodine intake, manage to compensate for these physiologic changes and remain euthyroid throughout pregnancy. Three percent to 5% of pregnant women have decreased thyroidal reserve and develop subclinical hypothyroidism (SCH) [8,9]. The latter, defined by serum thyroid-stimulating hormone (TSH) concentration greater than the pregnancy-specific reference range for each trimester [9], is more common than overt hypothyroidism. Mild thyroid dysfunction has been associated with impaired neuropsychological development of the offspring as well as adverse obstetric outcomes [2,10,11,12,13,14]. There is a consensus that SCH detected during pregnancy should be treated with levothyroxine (LT4), particularly in the presence of antithyroid antibodies [5,15,16,17]. Consequently, in an increasing number of women, LT4 therapy is started during pregnancy. However, there are limited data regarding the optimal management of these women postpartum: there are no relevant prospective studies and guidelines cover this issue only partially. The European Thyroid Association Guidelines focus solely on thyroid antibody-negative pregnant women started on LT4 for a TSH < 5 mIU/L. In these women, postpartum discontinuation of LT4 is recommended followed by TSH measurement after 6 weeks [16]. The 2017 American Thyroid Association Guidelines suggest postpartum discontinuation of therapy only in women who required LT4 ≤ 50 μg/day and reevaluation of serum TSH at 6 weeks postpartum [5].

To assess the necessity of continuation of LT4 replacement postpartum, we aimed to retrospectively study case files of women with SCH detected during pregnancy. Our working hypothesis was that in most cases this mild thyroid disorder might not resolve after delivery.

## 2. Materials and Methods

We conducted a retrospective cohort study, reviewing medical records of pregnant women with new onset SCH during pregnancy who attended our endocrinology unit between January 2017 and December 2018. We considered follow-up after delivery as being complete if each subject was assessed at 6 weeks, 6 months and 1 year postpartum. Women with pre-existing thyroid disease, past use of LT4 therapy, loss to follow-up after delivery and discontinuation of LT4 on their own volition or those who were non-compliant with medical recommendations were excluded. All the participants received daily multivitamin supplements during gestation and breastfeeding, including at least 140 mcg of iodine per day. A total of 114 women were eligible for analysis. The collected clinical data included demographic details such as age, gestational age at the time of diagnosis, primiparity, body weight gain during pregnancy, thyroid hormone requirement during pregnancy, presence of elevated antithyroid peroxidase antibodies (TPOAbs), thyroid ultrasound findings such as enlargement of the thyroid gland, inhomogeneity, hypoechogenicity, or diffusely increased vascularization of the thyroid gland (ultrasound performed with Hitachi Aloka F31 (Hitachi Aloka Medical Ltd., Tokyo, Japan)), or gestational diabetes, and mode of delivery. The decision to continue LT4 after delivery was taken based on the declared desire of women to breastfeed their neonates, serum TSH levels at the time of diagnosis higher than 5 mIU/L, or when daily dose of LT4 before delivery was more than 50 μg as well as when TPOAbs were positive [5,16,17,18,19,20]. Plasma TSH levels and TPOAbs were measured using electrochemiluminescent assays (Cobas Elecsys, Roche, Basel, Switzerland). All the samples were measured in duplicate; TPOAbs values >34 U/L were considered as being positive. SCH during pregnancy was defined as normal FT4 concentration and elevated serum TSH levels for the trimester-specific pregnancy reference range (more than 3.5 mIU/L in the first and subsequent trimesters), based on a more realistic approach of TSH cut-off (rather than 4.0 mIU/L), in order to comply to the newer 2017 ATA guidelines (and phase out simultaneously older criteria in our clinical practice) [5,15,16]. SCH after delivery was defined as elevated TSH levels (TSH > 4.5 mIU/L, the upper limit of the assay’s population reference range) with normal FT4 concentration (reference range: 9.2–19 pg/mL). Overt hypothyroidism was defined as elevated TSH with decreased FT4 levels [5,15,16]. The study was approved by our hospital’s scientific board/ethics committee (No 17/2019). Continuous variables data are presented as mean ± standard deviation (SD). Logistic regression was used to assess the risk of permanent hypothyroidism after delivery (statistical significance was set at *p* < 0.05, MedCalc statistical data software v.14.8.1 (MedCalc Software Ltd., Ostend, Belgium) was used).

## 3. Results

Details of the 114 women studied are given in Table 1 and Figure 1. Treatment with LT4 was initiated for mean ± SD serum TSH levels of 5.24 ± 2.55 mIU/L. The mean ± SD LT4 dose given was 77.9 ± 20.4 μg/day. Of the 114 women, 17 (14.9%) were TPOAbs (+), 21 (18.4%) had positive thyroid ultrasound findings and 32 (28.1%) developed gestational diabetes. Mean ± SD serum TSH levels before delivery were 1.65 ± 1.19 mIU/L with a mean ± SD adjusted daily treatment dose of 85.1 ± 23.2 μg. Following guidelines [5,16] and the criteria presented in the previous section, 98 patients were advised to continue LT4 after delivery on a lower dose (mean ± SD: 53.5 ± 28.1 μg/day). The remaining 16 women discontinued LT4 treatment.

On the first visit postpartum (6 weeks after delivery), mean TSH ± SD levels were 1.77 ± 2.1 mIU/L, and another 23 women stopped LT4 supplementation (followed by breastfeeding cessation). In the group of patients who discontinued LT4 right after delivery (16/114), 4 women restarted treatment due to SCH relapse (mean ± SD TSH levels were 8.48 ± 2.77 mIU/L). Thus, at 6 weeks postpartum, 79 women (69.3%) with new onset SCH during pregnancy were advised to continue the LT4 replacement therapy mainly because of exclusive breastfeeding. By the second visit (6 months postpartum), 28 women (24.5%) in total had stopped LT4 (7 patients of 23 women who discontinued LT4 at first postpartum visit restarted treatment with mean ± SD TSH levels at 9.44 ± 7.21 mIU/L). The mean ± SD daily LT4 dose in the remaining 86 (75.4%) who continued treatment was 50.1 ± 33.0 μg/day. At the final visit (1 year postpartum) most patients 86/114 (75.4%) were still on levothyroxine treatment. In 11 of the 39 women (28.2%) who had discontinued LT4 treatment postpartum, SCH relapsed (with mean ± SD TSH levels of 9.09 ± 5.81 mIU/L) and LT4 treatment was restarted.

Logistic regression (x^2^: 21.975, *p* = 0.0005) showed that the need for LT4 continuation postpartum was significantly related to TSH levels at the initiation of treatment in gestation (OR: 2.061 (95%CI: 1.259–3.373), *p* = 0.004) and inversely related to primiparity (OR: 0.330 (95%CI: 0.131–0.835), *p* = 0.019), but not related to age (*p* = 0.684), gestational age at the initiation of LT4 (*p* = 0.458), body weight change in pregnancy (*p* = 0.462), positive TPOAbs (*p* = 0.759), thyroid ultrasound findings (*p* = 0.2) or presence of gestational diabetes (*p* = 0.164). In the group of women who discontinued LT4 postpartum (39/114), the reinitiation of treatment due to SCH relapse was unrelated to age, gestational age at the initiation of LT4, initial serum TSH levels at diagnosis, body weight change, primiparity, TPOAbs presence or thyroid ultrasound findings (all *p* > 0.5).

## 4. Discussion

The present study was performed to evaluate the progress of new onset SCH in pregnancy and to elucidate the necessity of continuation of LT4 treatment postpartum. Sixty-nine percent of women in our study were advised to continue LT4 treatment 6 weeks postpartum, adhering to current guidelines (which suggest discontinuation of LT4 only if the daily dose is below 50 μg [5]); however, the mean daily LT4 in our study population was slightly higher (85.1 ± 23.2 μg). This fact was statistically related to TSH levels at the initiation of treatment in gestation, without any other associations to well-known factors in the literature such as positive TPOAbs. Furthermore, since breastfed infants depend exclusively on breast-milk iodine to meet their increased physiologic needs, and taking into consideration that iodine deficiency early in life can irreversibly impair neurodevelopment and maturation, we preferred to continue LT4 if neonates/infants remained exclusively breastfed (up to 6 months postpartum) [18,19,20]. In addition, up to 1-year postpartum, 11/39 (28.2%) women who discontinued LT4 developed SCH and LT4 therapy was restarted. Thus, most women who developed SCH during pregnancy continued or restarted LT4 treatment during lactation for up to 1-year postpartum. Our results lend credence (at least in part) to a significant number of previous studies. Ramesh et al. found that only 17.8% of women with SCH during pregnancy developed hypothyroidism up to 2 years postpartum [21]. In another recent study, Vaidya et al. showed that 75.4% of women with subclinical hypothyroidism in pregnancy had normal thyroid function post pregnancy at the 5-year follow-up [22]. However, Haddow et al. showed that 64% of women with undiagnosed mild hypothyroidism during pregnancy had a confirmed diagnosis of overt hypothyroidism on follow-up over a decade after delivery [10]. Nevertheless, their cohort had much higher TSH levels (mean TSH 13.2 mIU/L) than pregnant women in our study where the specific trimester reference ranges were used for the diagnosis of subclinical hypothyroidism [15,16].

We analyzed several possible predictive factors for the future risk of SCH and found no association of them with postpartum SCH relapse and need for LT4 reinitiation. The only significant association was between LT4 continuation postpartum with TSH levels at the initiation of treatment in gestation and inversely with primiparity, but not with reinitiation of treatment after discontinuation. This finding is in accordance—in part—with previous studies. Neelaveni et al. found that advanced age, goiter, positive family history of thyroid disease, higher LT4 requirement during the last trimester and thyroid autoimmunity increase the risk of hypothyroidism after delivery [21]. Moreover, they showed that higher initial TSH levels during pregnancy were predictive of future hypothyroidism relapse (81% of their pregnant women with initial TSH higher than 7.5 mIU/L developed hypothyroidism after delivery) [21]. In addition, Shields et al. found that a TSH concentration higher than 5 mIU/L in pregnancy was associated with increased risk for persistence of hypothyroidism after delivery [22]. The same study showed that TPOAbs (+) was the most important risk factor for future hypothyroidism [22]. Furthermore, in our study, there was a significant inverse association between primiparity and LT4 continuation postpartum, since we know that the number of previous pregnancies is a factor that affects thyroid autoimmunity and function [23].

There are limitations in our study due to its retrospective design based on an outpatient clinic, the short duration of the follow-up, and the relatively small number of the study group, compared to the prevalence of SCH in pregnancy, especially regarding the patients (11/39) in whom LT4 treatment was reinitiated postpartum (small number vis-à-vis underlying co-factors under investigation), and the relatively small number of TPOAb positive women (17/114). Plasma thyroglobulin antibodies (TGAbs) were also measured (values >115 IU/L were considered as being positive), but there was no statistically significant difference in the presence of TGAbs vs. TPOAbs in our study. Furthermore, the iodine status of the women in our study, was not specifically measured, but all the participants received at gestation and during breastfeeding, daily multivitamin supplements including at least 140 mcg of iodine per day.

In conclusion, an increasing number of women are diagnosed with new onset SCH in pregnancy. Since most women in our study, who were started on LT4 during pregnancy continued treatment postpartum, and a considerable number of those who had discontinued treatment, reinitiated LT4 after delivery, it is of great importance to assume that SCH diagnosed in pregnancy seems to be more complicated than it is expected to be postpartum. Women, who develop SCH during pregnancy and discontinue LT4 postpartum remain at increased risk of recurrence in a subsequent pregnancy; thus it is reasonable to assess thyroid function before conception or as soon as pregnancy is confirmed [22], and at 6 months to 1 year postpartum. However, further relevant research is essential to gather more robust evidence and to eventually formulate more specific guidelines.

## Figures and Tables

**Figure 1 medsci-08-00029-f001:**
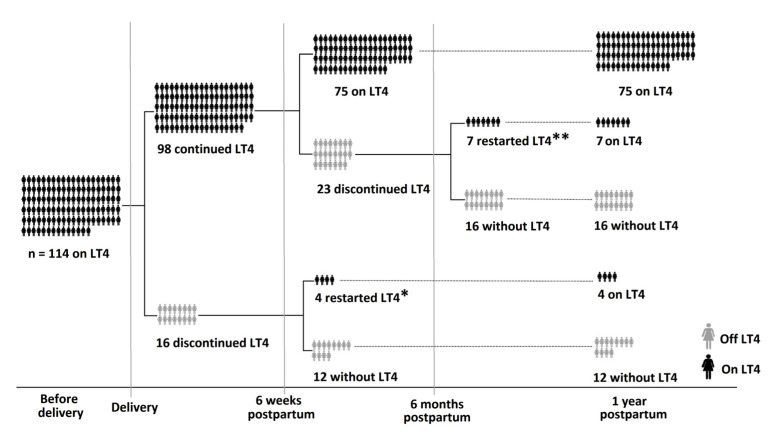
Flowchart of the 1-year postpartum follow-up. One year after delivery, most patients (86/114 (75.4%)) were still on levothyroxine treatment (LT4). In total, 11 out of 39 women who had discontinued LT4 after delivery, developed recurrence of subclinical hypothyroidism and reinitiated LT4. * In the first follow-up visit (6 weeks postpartum), 4 out of 16 women who had discontinued LT4 immediately after delivery, reinitiated it because of recurrence of hypothyroidism (mean ± SD TSH levels were 8.48 ± 2.77 mIU/L). ** In the second follow-up visit (6 months postpartum), another 7 out of 23 women who had discontinued LT4 after the first visit, reinitiated LT4 because of recurrence of hypothyroidism (mean ± SD TSH levels at 9.44 ± 7.21 mIU/L).

**Table 1 medsci-08-00029-t001:** Demographic characteristics, clinical data, and laboratory findings of the study group; * enlargement, inhomogeneity, hypoechogenicity, or diffusely increased vascularization of the thyroid gland.

N	114
Mean Age ± SD (years)	31.0 ± 6.1
Mean gestational age at diagnosis ± SD (weeks)	18.7 ± 8.6
Mean body weight gain during pregnancy ± SD (kg)	13.3 ± 5.5
Mean neonatal weight at delivery ± SD (gr)	3197.4 ± 535.8
Primipara	60/114 (52.6%)
Caesarean section	62/114 (54.3%)
Gestational Diabetes	32/114 (28.0%)
Positive TPOAbs during pregnancy	17/114 (14.9%)
Positive thyroid ultrasound findings *	21/114 (18.4%)
Mean TSH at diagnosis ± SD (mIU/L)	5.24 ± 2.55
Mean LT4 dose ± SD (μg/day)	77.9 ± 20.4
Mean TSH before delivery ± SD (mIU/L)	1.65 ± 1.19
Mean LT4 dose before delivery ± SD (μg/day)	85.1 ± 23.2

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
