# Peer review of "Postpartum Follow-Up of Women Who Developed Subclinical Hypothyroidism during Pregnancy"

_medsci, 2020, doi:10.3390/medsci8030029_

Round 1

Reviewer 1 Report

In this study Dr Linardi et al report on the postpartum use of L-t4 in women who developed subcl. hypothyroidism in pregnancy. This research is of clinical interest as much emphasis is usually given in the antenatal status of these women. The article is well-written. Nevertheless, it will benefit from a modest amount of revision as detailed below.

  • Introduction:
    • Re the definition of SCH you need to expand by stating that where there is no assay-specific ref range then a TSH up to 4 and 2.5miu/l may be considered as the upper limit of normal for TPOAb –ve and +ve patients, respectively (as per ATA 2017)
    • Your definition of SCH differs from the aforementioned definition – was there any specific reason why a TSH cut-off was chosen i.e. is this a locally agreed cut-off or do you have locally- and assay-specific and trimester-specific ref ranges?

  • Methods:
    • I note that all descriptive statistics are given as mean (SD). Where all your data normally distributed, even for analyses involving subpopulations? Please clarify
    • No info is given whether the various assumptions for logistic regression were tested and fulfilled
    • What statistical software did you use for your analyses?

  • Results:
    • For the 114 women who received L-t4 the TSH was 5.24+/-2.55 miu/l – does this mean some women were given L-t4 with a TSH < 3.5?
    • In table 1 I’d suggest adding percentages next to the fractions
    • Figure 1 very nicely illustrates the point you’re trying to convey
    • No need to keep repeating mean+/-SD; you stated that in your introduction

  • Discussion
    • Line 143 ‘as positive TPOAbs’ should be changed to ‘such as positive TPOAbs’
    • I would say the main limitations of your study is its retrospective nature with all the inherent difficulties of carrying out such research e.g. biases, difficulty in controlling confounding etc. Secondly, although the study nicely shows that most women carry on taking L-t4 a year after childbirth, it does not give us any info re hard outcomes i.e. how did the treated women benefited (health-wise) vs. those who did not? I suggest you mention these limitations. Finally, I assume that the iodine status of these ladies was not specifically measured, but this is worth mentioning as a limitation.

Author Response

Responses to comments

Reviewer 1 comments

“In this study Dr Linardi et al report on the postpartum use of L-t4 in women who developed subcl. hypothyroidism in pregnancy. This research is of clinical interest as much emphasis is usually given in the antenatal status of these women. The article is well-written. Nevertheless, it will benefit from a modest amount of revision as detailed below”

We appreciate the reviewer’s comment.

Introduction:

“Re the definition of SCH you need to expand by stating that where there is no assay-specific ref range then a TSH up to 4 and 2.5miu/l may be considered as the upper limit of normal for TPOAb –ve and +ve patients, respectively (as per ATA 2017)”

“Your definition of SCH differs from the aforementioned definition – was there any specific reason why a TSH cut-off was chosen i.e. is this a locally agreed cut-off or do you have locally- and assay-specific and trimester-specific ref ranges?”

In order to comply with newer guidelines from ATA (and phasing out simultaneously older criteria in our practice), we used a more realistic approach in TSH cut-off, based on our previous experience as a reference center for endocrinopathies in pregnancy, and a well-established trimester-specific reference range from our Endocrine laboratory. In response to these comments we rephrased the text to be more concise as follows: “…SCH during pregnancy was defined as normal FT4 concentration and elevated serum TSH levels for the trimester-specific pregnancy reference range (more than 3.5 mIU/L in the first and subsequent trimesters), based on a more realistic approach of TSH cut-off (rather than 4.0 mIU/L), in order to comply to the newer 2017 ATA guidelines (and phase out simultaneously older criteria in our clinical practice) [5,15,16]…” line 81-85

Methods:

“I note that all descriptive statistics are given as mean (SD). Where all your data normally distributed, even for analyses involving subpopulations? Please clarify”

“No info is given whether the various assumptions for logistic regression were tested and fulfilled”

We performed Logistic regression; therefore, it does not require a linear relationship between the dependent and independent variables. The residuals do not need to be normally distributed, and homoscedasticity is not required. Parametric assumptions like normality and homoscedasticity are not relevant in logistic regression. Finally, the dependent variable in logistic regression is not measured on an interval or ratio scale. We used mean values for better visualization of our data.

[Osborne, J. (2015). A practical guide to testing assumptions and cleaning data for logistic regression. In Osborne, J. Best practices in logistic regression (pp. 84-130). 55 City Road, London: SAGE Publications, Ltd doi: 10.4135/9781483399041]

“What statistical software did you use for your analyses?”

MedCalc statistical data software v.14.8.1 (MedCalc Software Ltd, Ostend, Belgium) line 91-92

Results:

“For the 114 women who received L-t4 the TSH was 5.24+/-2.55 miu/l – does this mean some women were given L-t4 with a TSH < 3.5?”

The corresponding median TSH value was 4,53 mIU/L. These are the SDs, not the upper and lower limits.

“In table 1 I’d suggest adding percentages next to the fractions”

We added percentages in table 1 per suggestion.

“Figure 1 very nicely illustrates the point you’re trying to convey”

We appreciate the reviewer’s comment.

“No need to keep repeating mean+/-SD; you stated that in your introduction”

This was done to facilitate readability with data scattered through the text.

Discussion

“Line 143 ‘as positive TPOAbs’ should be changed to ‘such as positive TPOAbs”

We rephrased this per suggestion. line 148

“I would say the main limitations of your study is its retrospective nature with all the inherent difficulties of carrying out such research e.g. biases, difficulty in controlling confounding etc. Secondly, although the study nicely shows that most women carry on taking L-t4 a year after childbirth, it does not give us any info re hard outcomes i.e. how did the treated women benefited (health-wise) vs. those who did not? I suggest you mention these limitations. Finally, I assume that the iodine status of these ladies was not specifically measured, but this is worth mentioning as a limitation.”

This was a retrospective study based on an outpatient clinic. The iodine status was difficult to measure in an everyday clinical practice. However, all women in our study, received daily multivitamin supplements including at least 140 mcg/day of iodine. We rephrased the text to be more concise as follows: “…All the participants received during gestation and breastfeeding, daily multivitamin supplements including at least 140 mcg of iodine per day…” line 67-69

“…Furthermore, the iodine status of the women in our study, was not specifically measured, but all the participants received at gestation and during breastfeeding, daily multivitamin supplements including at least 140 mcg of iodine per day…” line 188-190

Reviewer 2 Report

This study assessed the necessity of continuation of LT4 therapy in a cohort of pregnant women. Authors found that TSH levels at the initiation of treatment in gestation was directly related to the need for LT4 continuation, while primiparity was inversely related.

The article has a potential interest but there are some points that need to be clarified.

  • Any data about TGAb?
  • In patients who had a relapse of SCH, did the author exclude the presence of post partum thyroiditis?
  • It is not clear to me the rationale to continue LT4 therapy if neonates/infants remained exclusively breastfed. In line 1444-147, Authors reported that iodine deficiency early in life can irreversibly impair neurodevelopment and maturation, that is true. But to avoid this complication, one should suggest iodine supplement instead hormone therapy.
  • Logistic regression revealed the lack of association of TPOAb, a marker thyroid autoimmunity. Other than the absence of TGAb, which reduces the frequency of thyroid autoimmunity, another reason is the low number of TPOAb positive women (17/114). Authors should add this as a limitation of the study because thyroid autoimmunity is one of the most important risk factor for the development of hypothyroidism in all patients, pregnant women included.

Author Response

Reviewer 2 comments

This study assessed the necessity of continuation of LT4 therapy in a cohort of pregnant women. Authors found that TSH levels at the initiation of treatment in gestation was directly related to the need for LT4 continuation, while primiparity was inversely related.

The article has a potential interest but there are some points that need to be clarified.

 We appreciate the reviewer’s comment.

“Any data about TGAb?”

There was no difference, in our study, regarding TGAb positivity vs TPOAb, therefore we included only TPOAb presence as a variable since most of the literature for subclinical hypothyroidism in pregnancy reports TPOAb presence. We rephrased the text to be more concise as follows: “…Plasma Thyroglobulin antibodies (TGAbs) were also measured (values >115 IU/L were considered as being positive), but there was no statistically significant difference in the presence of TGAbs vs TPOAbs in our study…” line 185-188

“In patients who had a relapse of SCH, did the author exclude the presence of postpartum thyroiditis?”

We did measure thyroid antibodies, but we cannot exclude postpartum thyroiditis since subclinical Hypothyroidism might be related to that.

“It is not clear to me the rationale to continue LT4 therapy if neonates/infants remained exclusively breastfed. In line 1444-147, Authors reported that iodine deficiency early in life can irreversibly impair neurodevelopment and maturation, that is true. But to avoid this complication, one should suggest iodine supplement instead hormone therapy.”

Based on our endocrine outpatient clinic experience, all women in our study received daily multivitamin supplements including at least 140 mcg/day of iodine, in gestation and most of them during breastfeeding. Levothyroxine treatment, whenever indicated due to increased levels of TSH, should eventually increase daily iodine status. We rephrased the text to be more concise as follows: “…All the participants received during gestation and breastfeeding, daily multivitamin supplements including at least 140 mcg of iodine per day…” line 67-69

“…Furthermore, the iodine status of the women in our study, was not specifically measured, but all the participants received at gestation and during breastfeeding, daily multivitamin supplements including at least 140 mcg of iodine per day…” line 188-190

“Logistic regression revealed the lack of association of TPOAb, a marker thyroid autoimmunity. Other than the absence of TGAb, which reduces the frequency of thyroid autoimmunity, another reason is the low number of TPOAb positive women (17/114). Authors should add this as a limitation of the study because thyroid autoimmunity is one of the most important risk factor for the development of hypothyroidism in all patients, pregnant women included.”

We rephrased the text to be more concise as follows: “…Plasma Thyroglobulin antibodies (TGAbs) were also measured (values >115 IU/L were considered as being positive), but there was no statistically significant difference in the presence of TGAbs vs TPOAbs in our study…” line 185-188